# The Executive-Function-Related Cognitive–Motor Dual Task Walking Performance and Task Prioritizing Effect on People with Parkinson’s Disease

**DOI:** 10.3390/healthcare11040567

**Published:** 2023-02-14

**Authors:** Yen-Po Lin, I-I Lin, Wei-Da Chiou, Hsiu-Chen Chang, Rou-Shayn Chen, Chin-Song Lu, Ya-Ju Chang

**Affiliations:** 1Department of Medical Education, National Cheng Kung University Hospital, Tainan 704, Taiwan; 2Department of Medical Education, Kaohsiung Veterans General Hospital, Kaohsiung 813, Taiwan; 3School of Physical Therapy and Graduate Institute of Rehabilitation Science, College of Medicine, Chang Gung University, Taoyuan 333, Taiwan; 4Department of Physical Rehabilitation, Kaohsiung Armed Forces General Hospital, Kaohsiung 802, Taiwan; 5Professor Lu Neurological Clinic, Taoyuan 333, Taiwan; 6Department of Neurology, Chang Gung Memorial Hospital Linkou, Taoyuan 333, Taiwan; 7School of Medicine, College of Medicine, Chang Gung University, Taoyuan 333, Taiwan; 8Neuroscience Research Center, Chang Gung Memorial Hospital Linkou, Taoyuan 333, Taiwan; 9Health Aging Research Center, Chang Gung University, Taoyuan 333, Taiwan

**Keywords:** Parkinson’s disease, dual task, walking, Stroops, cognitive test, spatial memory

## Abstract

To safely walk in a community environment requires dual cognitive–walking ambulation ability for people with Parkinson’s disease (PD). A past study showed inconsistent results on cognitive–walking performance for PD patients, possibly due to the various cognitive tasks used and task priority assignment. This study designed cognitive–walking tests that used executive-related cognitive tasks to evaluate patients with early-stage Parkinson’s disease who did not have obvious cognitive deficits. The effect of assigning task prioritization was also evaluated. Sixteen individuals with PD (PD group) and 16 individuals without PD (control group) underwent single cognitive tests, single walking tests, dual walking tests, and prioritizing task tests. Three types of cognitive, spatial memory, Stroops, and calculation tasks were employed. The cognitive performance was evaluated by response time, accuracy, and speed–accuracy trade off composite score. The walking performance was evaluated by the temporal spatial gait characteristics and variation in gait. The results showed that the walking performance of the PD group was significantly worse than the control group in both single and dual walking conditions. The group difference in cognitive performance was shown in composite score under the dual calculation walking task but not under the single task. While assigning priority to walking, no group difference in walking was observed but the response accuracy rate of PD groups declined. This study concluded that the dual task walking test could sharpen the cognitive deficits for early-stage PD patients. The task priority assignment might not be recommended while testing gait deficits since it decreased the ability to discriminate group differences.

## 1. Introduction

Parkinson’s disease (PD) is the second-most-common neurodegenerative disease, and the proportion of the population suffering from Parkinson’s disease increases with age [1]. The prevalence of PD is around 0.3% of the entire population and about 1% in people over 60 years of age, but the numbers vary according to different geographical area and methodology of investigation [2,3]. Generally, the average age of onset is between 50 and 60 years old [3]. Insufficient dopamine secretion can also affect the frontostriatal circuits, which causes cognitive problems [4,5]. Clinical diagnosis relies on cardinal motor features, but non-motor symptoms, such as cognitive deficits, could increase the overall disability [6]. Declined walking ability is an important deficit that causes falls and prevents patients from outdoor functional activities and, thus, affects patients’ quality of life [7]. Precise evaluation of the patients’ walking ability, especially the walking ability that could reflect their community walking ability, is important for clinical decision making in terms of adjusting medication and rehabilitation plans [8]. It is not easy to evaluate community ambulation ability clinically. People often perform walking under cognitive load (dual walking tasks) in their daily lives, such as memorizing shopping lists while walking, calculations while walking, and so on. Walking under cognitive load requires simultaneous control of walking and cognitive functions [9]. It has been demonstrated that even young healthy adults walk slower when they are required to walk while performing another task [10]. This paradigm is also known as the dual task walking paradigm. Under the execution of one task, the performance of another task in progress may be poor. In this paradigm, a deficit could be shown in gait, such as a reduction in walking speed. Reductions in step length and increases in the coefficient of variation (CV) of walking speed and step length, or cognitive performance, such as a decrease in the correct response rate and an increase in response time, are also known as deficits in the dual walking task paradigm [11,12,13,14].

Compared with healthy people of the same age, studies have shown that patients with Parkinson’s disease are more susceptible to walking deficits under cognitive load [12], which was related to a decrease in gait automaticity. The dual task walking test has been popular clinically due to this paradigm, which is suggested to make the gait automaticity deficit more obvious to detect under cognitive loads. In a previous study, Zirek et al., compared 48 PD and 48 control subjects to analyze the effect of cognitive activities on the walking ability of patients [15]. The result showed that the completion time of a quick test for the assessment of mobility, balance, and risk of falls in the PD group was longer than the healthy controls under cognitive loading conditions.

Similar results were also found in the dual task during a motor cognitive task experiment by Johansson et al., a total of 93 patients with PD showed significantly worse postural control and pace when walking with cognitive load versus no cognitive load [16]. The hypothesized mechanism is that the PD patients use more cortical attention resources than healthy people to control gait [12,17]. Therefore, as the patient is performing a dual walking task, the two tasks compete for the limited attention resources of the cerebral cortex, resulting in a lack of ability to execute dual walking tasks [18,19]. The dual task test paradigm could be explained by capacity theory and/or bottleneck theory [12,20]. Capacity theory holds that attention capacity is limited, but the brain can flexibly allocate attention to specific tasks autonomously. If both tasks require attention, because the attention capacity is limited or the attention cannot be divided appropriately, as in patients with Parkinson’s disease, the dual walking task deficit will be shown [21]. The bottleneck theory believes that when two tasks pass through the same processing path, the center will process one task first and delay the processing or reduce the performance of the other task [21,22,23].

Although the cognitive–walking paradigm has been suggested for evaluating gait automaticity, past results of the dual task walking test on PD patients are inconsistent. In a study designed by Salazar et al., the variability in motor performance remained unaffected under the dual task that consisted of walking while performing the oral trail making test [24]. However, for 30 PD patients in another study when walking under the dual task condition, the subjects’ speed was greatly reduced and stride time variability became very large [25]. The inconsistent results could be due to different dual task designs and evaluation parameters used in different study designs. For example, different cognitive loads may apply to different brain regions and result in different assessments. In addition, factors, such as disease severity, bradykinesia, rigidity, postural instability, and other factors, can lead to reduced walking control. Slow movement results in reduced step length and slower walking speed [26]. Impaired executive function, attention, processing speed, memory, and visual space can also cause a loss of dual walking task ability [27,28]. The executive function of patients with Parkinson’s disease is significantly impaired [9,29], especially in the aspect of response inhibition and spatial memory [30,31]. Previous studies on the evaluation of dual task walking ability in PD patients did not employ a cognitive function that focused on executive function. It is possible that that a dual task paradigm requiring walking under executive-function-related cognitive load might enhance the motor deficit and make the test more sensitive for early-stage PD patients. However, there is no previous study design on the cognitive–walking dual task walking paradigm for PD patients, especially for early-stage patients.

Another potential factor to influence the application of the dual task walking test is the task prioritization assignment. Researchers [32] reported that healthy people prioritize gait stability over cognitive tasks when no specific prioritization instruction or allocation of attention is given to prevent a loss of balance. Studies also showed that patients with PD prioritized the cognitive task when they did not receive explicit instructions to prioritize gait, which inadvertently increased their risk for falling [33]. However, other studies quantified both primary and secondary tasks and found that the dual task did not always cause interference. A dual task facilitation phenomenon was also reported, but not systematically, in previous studies [34]. This condition cannot be explained by the conventional capacity or bottleneck models. The prediction of traditional models is worse in conditions when the prioritization was assigned. Another limitation of the dual task walking test for PD patients might come from the influence of the prioritization of task. O’Shea et al., compared 15 patients with PD and 15 healthy adults and reported that focus on walking improved the walking speed but cognitive task performance reduced in PD [21]. This is a phenomenon that can be predicted by the dual task theories. However, Yogev-Seligmann’s study, which used a verbal fluency walking paradigm to test young and old people [35], found that gait speed was reduced when priority was given to the cognitive task, but the effect was less dramatic in older adults. This suggested that when priorities are assigned, older people do not use allocation strategies as young people do. In addition, dual task costs related to prioritization were not associated with the baseline executive or cognitive function in older adults. In a recent study by Longhurst et al. [34], in which they evaluated the dual task effect separately on cognitive and motor tasks, they found the prioritization category was not reliable.

The primary aim of this study is to evaluate whether the executive-function-related cognitive–walking test paradigms could reveal deficits, which could not be shown in the single task paradigm in early-stage PD patients. We hypothesize that this paradigm will show significant differences in gait (speed, step length, and cadence) and/or cognitive functions (response time and accuracy) between the early-stage PD patients and control group, which could not be shown in the single walking or cognitive tests. The secondary aim is to evaluate the effect of assigning prioritization on the executive-function-related cognitive–walking test paradigms.

## 2. Materials and Methods

### 2.1. Research Design and Subjects

This study is a case-control study. Participants were 16 patients with PD (PD group) and 16 healthy subjects (control group). One subject in the control group withdrew due to personal reasons. Patients with Parkinson’s were recruited from the Department of Neurology division at a medical center. Age-matched control subjects with an average age of 60.93 ± 2.87 were recruited from the community. Parkinson’s patients must meet the following admission and inclusion criteria: idiopathic Parkinson’s disease with Hoehn and Yahr stage (H&Y) 1–3 (mild to moderate unilateral or bilateral disease; some postural instability; physically independent) levels diagnosed according to the United Kingdom Brain Bank Criteria. All PD participants had the postural instability gait difficulty (PIGD) motor subtype, without freezing of gait and with stable medication usage. Stable medication usage refers to no medication dosage changes three months before enrolling in the study and during the test period. Patients had physical/mental adaptations with effective maintenance by medication without symptom progress. Eligibility for inclusion was assessed by a neurologist. Subjects in both groups must be able to walk at least six meters indoors without using assistive devices and score at least 26 in Montreal Cognitive Assessment (MoCA) to avoid the influence of mild cognitive impairment [36,37]. The excluded conditions for both groups are: diagnosis of central nervous system diseases other than Parkinson’s disease, cardiopulmonary vascular disease, orthopedic disease, audiovisual sensory disease, symptoms of dementia, dyskinesia, or receiving deep brain stimulation. Subjects in the PD group were tested clinically “on” medication state. The single and dual task evaluations were performed by trained research team members who are licensed physical/occupational therapists. Patients and assessors were blind to the purpose of the study. Only the patients’ name and age were known prior to the subject’s evaluation; details were unknown. This study was approved by the Institutional Review Board and informed consent was obtained.

After inclusion, subjects were assessed by the MoCA, which is a scale for quick screening of mild cognitive impairment. This assessment was suggested to be sensitive in screening patients with early cognitive impairment [38]. By definition, a score of ≥26 points is considered normal.

Subjects then received a single cognitive test, single walking test, and dual walking test. The total duration of evaluation, including rest time, was less than 1.5 h to avoid fatigue. To avoid order effects, the experimental procedure was to perform single and dual tasks twice each. The three cognitive tasks were randomly assigned. The contents of cognitive questions for the same paradigm (spatial memory, Stroops, or calculation) were different between the two tests. Between tasks, there were at least two minutes for subjects to rest. The average of the two measurements was used for analysis. The single cognitive test was tested in a sitting position. Wireless headset microphone (Aibo, Mini A8, LY-MIC-BTA8, New Taipei City, Taiwan) was used to record the oral response to the cognitive tasks. The audio signal was sent to a computer through Bluetooth wireless transmission (v4.0) so the response time could be accurately analyzed corresponding to the test question. There are three cognitive tasks used in this study: spatial memory task, visual Stroop task, and calculation task. Each task is performed twice. The spatial memory task is to memorize the position and order of three yellow squares in a blank nine-square grid. Each position is given 1 s of memory time. Then, the yellow squares disappear and the numbers are shown on the nine-square grid. Subjects had to say the three corresponding numbers in the nine-square grid. For the visual Stroop task, there are colorful words shown on screen with one word each time. Subjects had to name the color of the word. For example, for a word “purple” shown in red, the answer should be red instead of purple. The colors and the words were all incongruent to increase the test difficulty. When performing this task, automated responses must be suppressed, which is used to assess response inhibition [39]. The calculation task required serial subtraction of “three” beginning with a number randomly selected between 60 and 100, which is a test commonly used in dual task tests in patients with Parkinson’s disease [40]. Before the three cognitive tasks officially started, there were 10 sample questions each for subjects to practice to ensure they understood the test.

The single walking task requires subjects to walk on a pressure mat (GAITRite, CIR systems Inc, Franklin, NJ, USA) (Figure 1) at a self-chosen comfortable speed and at the fastest speed. Each task is performed twice. When the subject walks across the trail, the temporal spatial parameters of gait, step time, step length, speed, and coefficient of variation (CV) were recorded.

For the dual walking task, subjects walked at a self-chosen comfortable speed and performed cognitive tasks simultaneously. Three combinations of dual tasks were employed in this study: walking combined with spatial memory tasks, walking combined with visual Stroop tasks, and walking combined with calculation tasks. In this dual walking test, no task prioritization was assigned.

Following dual walking test, subjects received dual calculation walking tests with priority assigned. While priority was assigned to the cognitive task, subjects were instructed to focus their attention on calculation rather than walking. In contrast, while priority was assigned to the walking task, subjects were instructed to focus their attention on walking rather than calculation. The research flowchart and experimental environment settings are as shown in Figure 1 and Figure 2, respectively.

### 2.2. Data Processing and Analysis

Performance parameters include walking speed (cm/s), step length (cm), CV of step length and step time (%), the response accuracy rate (%), reaction time (milliseconds), and composite score (%) of cognitive tasks.

The CV of response accuracy and reaction time was calculated by the following formulas:CV = (Standard Deviation/Mean) × 100%(1)
Response accuracy = number of correct responses total number of stimuli × 100%(2)
Composite score = response accuracy/reaction time × 100%(3)

### 2.3. Statistical Analysis

SAS (version 9.4; SAS Institute, Cary, NC, USA) was used for statistical analysis. Because some of the parameters show a non-normal distribution (response accuracy), the non-parametric test is used. The Chi-square test compares the ratio of males to females in the PD and healthy groups. The Mann–Whitney U test was used for between-group comparison and the Wilcoxon signed-rank test was used for within group comparison. The significance level was set at *p* < 0.05.

## 3. Results

### 3.1. Patient Characteristics

In total, 31 subjects were analyzed, 16 in the Parkinson’s disease group (5 female, 11 male, age: 63.13 ± 5.75 years) and 15 in the control group (4 female, 11 male, age: 60.93 ± 2.87 years). There was no statistically significant difference between the Parkinson’s and control group in the basic data (Table 1). Clinical characteristics of the participants, such as leg length, education level, MoCA, modified H&Y, disease duration (yrs), and levodopa equivalent dosage (mg/day), are shown in Table 2. The average levodopa dosage was 553.75 mg/day.

### 3.2. Gait Performance of Healthy Group and Parkinson’s Disease Group under Single Walking Task and Dual Task

In Figure 3, the Mann–Whitney U test showed that the walking speed of the Parkinson’s disease group was significantly lower than the control group in the single walking task (*p* = 0.005), dual spatial memory task (*p* = 0.002), dual Stroop Task (*p* = 0.044), and dual calculation task (*p* = 0.009).

In terms of the step length performance, the Parkinson’s disease group showed significantly shorter step length in the single walking task (*p* = 0.003), dual walking with spatial memory task (*p* = 0.003), dual walking with Stroop task (*p* = 0.027), and dual walking with calculation task (*p* = 0.003) than the healthy group.

In the CV of step length, the PD group showed greater CV than the control group in only the dual walking with spatial memory task (*p* = 0.024) and Stroop task (*p* = 0.010), showing no difference in the single walking task (*p* = 0.155).

In terms of step time, no significant difference was found between the groups in the single walking task or any dual task conditions (*p* > 0.05)

The CV of step time in the PD group was greater than the control group in the single walking task (*p* = 0.022). No difference in CV was observed in the dual spatial memory task (*p* = 0.843), dual Stroop task (*p* = 0.553), and dual calculation task (*p* = 0.782) in these two groups.

### 3.3. Cognitive Performance of Healthy Group and PD Group under Single Cognitive Task and Dual Task

For response accuracy and reaction time, no group differences were found between PD and control groups in the single or any of the dual cognitive tasks (Figure 4). These suggested that the subjects in this study did not have obvious impairments in cognitive function.

In terms of the composite score, there was no significant difference between the groups in space memory task or Stroop task under the single and dual task conditions (*p* > 0.05). However, in the dual calculation task, the composite score in the PD group was significantly lower than that in the control group (*p* = 0.018) (Figure 4). This suggests that the PD patients had disadvantaged cognitive function under the dual calculation task.

### 3.4. The Performance of Dual Calculation Tasks in the Control Group and the Parkinson’s Disease Group under Designated Task Priority (Comparing the Difference between Gait Priority and Calculation Priority)

When priority was assigned to the walking task, no group difference was found. The PD group showed lower response accuracy than the control group (control: 95.33 ± 13.56%, PD: 72.50 ± 32.51% *p* = 0.037) but not the reaction time (*p* = 0.651) or composite score (*p* = 0.156). This suggested an improvement in gait performance with a detriment in cognitive performance. When priority was assigned to cognitive tasks, no group difference was observed in either walking performance or cognitive performance (Table 3).

## 4. Discussion

In this study, the walking performance in the early Parkinson’s disease group (speed, step length) was significantly worse than the healthy group, regardless of the single task or dual task walking. The step length CV was not different between groups in the single walking task but was different in dual walking. In other words, the dual walking with spatial memory and Stroop task sharpened the deficits of step length CV. The step time CV showed group differences in single task walking but the difference was gone (no difference/reduced to insignificant level) in the dual task walking due to the increase in step time CV in the healthy group. The cognitive performance (accuracy rate, reaction time) showed no between-group difference if tested in the single task condition. The group difference was shown in composite score under the dual calculation walking task. These results suggested that the dual task paradigm sharpened the cognitive test and step length CV and saturated the step time CV for early-stage PD patients. Furthermore, the results suggested that, in the PD group, when the focus was prioritized on walking, group differences in walking parameters were gone (no difference/reduced to insignificant level) with a lower response accuracy rate. In contrast, when the focus was prioritized on the cognitive task, the between-group difference on walking was gone, with no detriment on cognitive tasks.

The walking speed for a single task was less in the PD group compared to the control group, regardless of single or dual task conditions. Past studies showed that the speed of a single walking task was affected by disease, age, test environment, and related gait parameters [41]. The slower walking speed in the PD group was shown by a shorter step length but not the step time [42]. This finding was consistent with previous studies, which reported that slow movement in PD patients was related to reductions in speed and step length [12,41,43]. The patients in our study were relatively early-stage patients whose disease duration was around 5.25 years. Our results showed that, even in the early stage of PD, walking speed significantly declined, mainly due to the decline in step length. This was in accordance with previous studies. Short step length could be related to the reduced balance ability, in that people with impaired balance usually had short step length, according to the reports [44]. Our study showed that the single limb support ability that was reduced in the PD group provided evidence that the decline in walking speed in the PD group was related to balance function rather than dual-task-related detriments.

Some previous studies suggested that the variability in walking was a significant feature in PD patients [45]. Our study showed that the PD group had more CV in step time but not step length in single walking if tested in the single task condition. The increased step time CV in the PD group was attributed to the declined automaticity [46]. However, the group difference of step time CV was gone in the dual walking condition. A closer look into the data showed that this was caused by an increase in step time CV in the control group in the dual task condition and, possibly, resulted in the celling effect. Dual task walking has been suggested to trigger the automaticity of gait [47]. However, it might still require some subconscious resources, such as cerebellar-related gait coordination control, to regulate the step time [48]. The dual task walking might occupy these resources and increased step time CV in the control group, whereas these resources had already been occupied during single walking in the PD group.

The performance of dual task walking has been suggested to show the ability of walking automation [49]. In our study, we employed three types of cognitive task, spatial memory, visual Stroop, and calculation tasks, as the dual task paradigms. Both groups showed significant decreases in walking speed and step length under all three types of dual task walking conditions, and the step time was insensitive to different conditions. In general, visual Stroop tests are included in executive function tasks and correlate with the functioning of frontal regions [50]. Compared to the visual Stroop task, spatial memory engages different visual selection, both of which are required in normal attention processing [51]. In the study of Ambrosini et al., they found that participants with stronger resting-state-related activity in the left prefrontal region were better able to solve spatial tasks than the visual Stroop task [52]. However, evoked potentials in the left temporal region reduced during arithmetic tasks [53]. According to previous studies, the spatial memory task requires short-term memory, memory encoding, and retrieval processes [54], and the calculation task requires working memory and performing mental manipulation [55]. These different tasks were considered to affect the extent of cognitive performance, and different levels of difficulties will also cause different levels of interference [56]. Further studies are required for the detailed mechanism. In our study, a possible reason that all three types of dual walking paradigms had similar effects on walking parameters might be that the cognitive tasks employed were related to the executive function. No previous research has compared the walking performance of the Parkinson’s disease group and the healthy group on dual spatial memory or dual visual Stroop tasks. Wild et al. [57] compared walking with an executive-function-related calculation task, and non-related phoneme monitoring and story content comprehension. They showed that PD patients’ walking speed significantly slowed down only in the calculation conditions, suggesting that the executive function task was more sensitive than the non-executive function task in detecting walking deficits in PD patients.

In terms of cognitive variables, the results of our study showed no group difference in the accuracy or reaction time in single spatial memory, Stroop task, calculation task, and dual task if no prioritization was assigned. In the single task condition, both the control and the PD group had at least 90% accuracy. There was also no significant group difference in education level or Montreal Cognitive Assessment. Our results supported that the patients in the PD group had no obvious cognitive functional decline. Researchers suggested that cognitive dysfunction is independent of motor disability [58]. Wild et al., reported non-significant results in the correct number of single and dual consecutive minus seven calculation tasks in a healthy and PD group [57]. Our study showed that if a composite score was employed in a dual task paradigm, the group difference could be shown in a dual calculation task. This suggested that the dual task paradigm using a composite score sharpened the cognitive test for early-stage PD patients. Fitts et al., proposed a speed–accuracy phenomenon [59]; that is, measurement of the single parameters might cause variability and decrease the sensitivity of the test. The composite score of speed and accuracy could overcome the speed–accuracy phenomenon and could be a sensitive cognitive indicator, especially for early-stage PD.

Parkinson’s disease (PD) can be classified into different subtypes, such as PIGD, tremor dominant, and indeterminate [60]. These subtypes are classified according to their crucial feature of locomotor functions, which affected their risk of falls [61]. A recent study conducted by Onder et al., found that the completion time of a time-up-and-go task (TUG) under a cognitive dual task condition was positively correlated with the UPDRS motor scores and FOG scores [62]. Therefore, it is reasonable to suspect that the results of a cognitive–walking test would be influenced by different subtypes of PD patients. However, the subtypes did not influence the results of our study. This may be due to the fact that all the patients enrolled in our study were early-stage PIGD subtypes. A detailed discussion of the influence of subtypes is suggested in a future study with a different design.

The results of our study showed that with the task priority assigned to either walking or cognitive tasks, the between-group difference in walking performance disappeared. Previous studies reported that a gait prioritization strategy could ameliorate gait deficits immediately [63,64] and after short-term training [65]. Our study further showed that, in early-stage PIGD patients, prioritizing attention diminishes the difference between PD and control groups. This could be explained by the task prioritization ability not being impaired in PD patients. Yogev-Seligmann et al., studied PD patients with advanced motor symptoms and found the task prioritization abilities were similar to non-PD controls. They suggested that PD patients without significant cognitive impairment could utilize their cognitive resources in the same manner as healthy comparisons [66]. Therefore, prioritizing attention to gait, or posture-first strategy, could be used clinically to overcome gait deficits but is not appropriate in discriminating gait deficits from PD to non-PD groups. While assigning task priority to walking, the PD group showed lower response accuracy than the PD group. The phenomenon could be explained by the capacity sharing model that PD subjects had limited total capacity to allocate attention to the secondary task, leading to a decline in the secondary task. On the contrary, when priority was assigned to cognitive tasks, the group differences in both walking and cognitive performance were insignificant and the walking performance was worse than that while priority was assigned to walking. This phenomenon suggests that assigning priority to cognitive is inappropriate for both PD and non-PD people.

Our approach has some inherent potential limitations. First, the PD patients enrolled in the study had the PIGD subtype and were in the early stage, so the results could not be inferred for other subtypes or different stages (ex. preclinical or advanced stage). Whether the dual cognitive–walking paradigm is essential to distinguish preclinical PD patients requires further study. Second, patients with PD were evaluated under the clinical “ON” state; whether the drug withdrawal state has a similar impact remains to be investigated. Third, patients with PD are prone to fatigue [67]. Although this study could not fully control the fatigue factors, rest periods were given during the test, and all subjects had the same test sequence in each assessment so that the test results could be compared at different times. Thus, we believe that the risk of bias was small.

## 5. Conclusions

In conclusion, our study showed that gait performance in the single walking task and the dual task can distinguish the healthy group from the early-stage Parkinson’s disease group. The dual task paradigm, considering the speed–accuracy trade-off, could sharpen the cognitive test and detect the cognitive deficits in early-stage PD patients. Moreover, the task priority assignment might not be recommended while testing gait deficits for early-stage PD patients.

## Figures and Tables

**Figure 1 healthcare-11-00567-f001:**
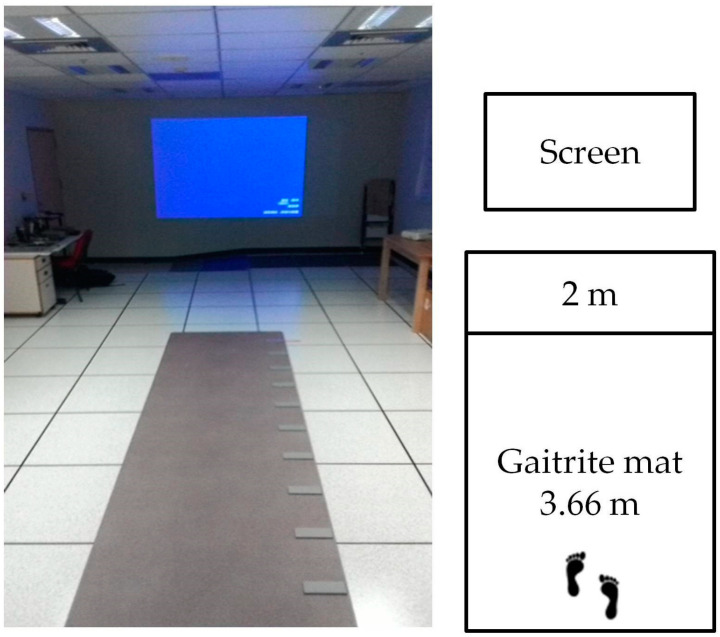
Experimental environment settings.

**Figure 2 healthcare-11-00567-f002:**
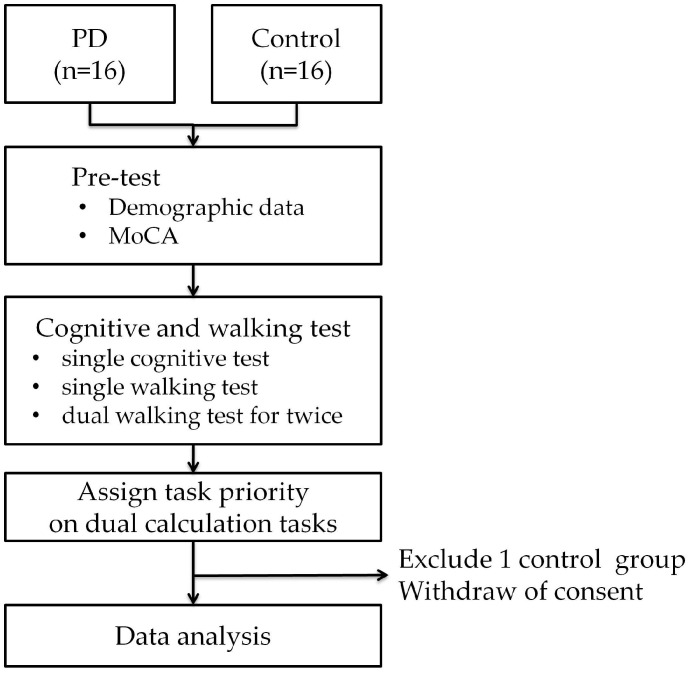
Flowchart.

**Figure 3 healthcare-11-00567-f003:**
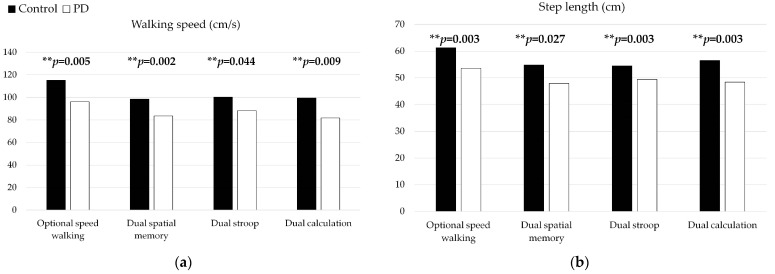
The walking performance includes (**a**) Walking speed, (**b**) Step length, (**c**) Step time, (**d**) CV Step length, and (**e**) CV Step time, under the group single and dual tasks between healthy and PD group. ** *p*< 0.05.

**Figure 4 healthcare-11-00567-f004:**
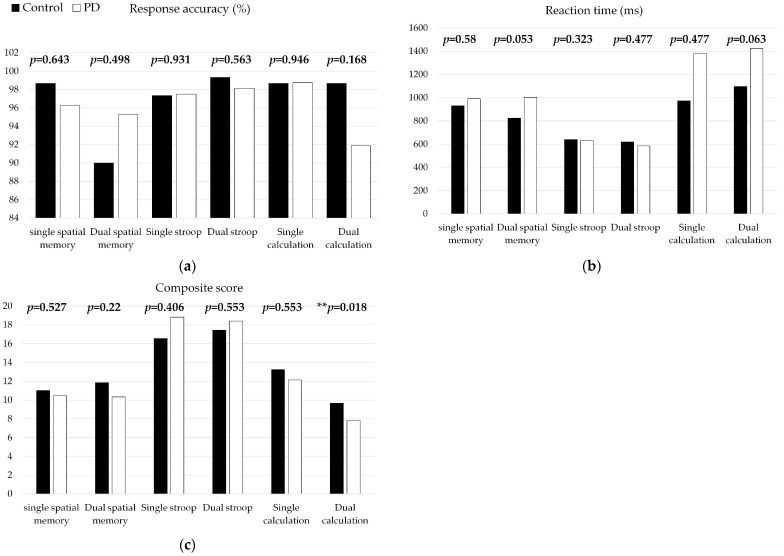
The cognitive performance includes (**a**) Response accuracy, (**b**) Reaction time, and (**c**) composite score, under the group single and dual tasks between healthy and PD group. ** *p* < 0.05.

**Table 1 healthcare-11-00567-t001:** Characteristics of the 31 patients with or without Parkinson’s.

Variable	Healthy Group (*n* = 15)	PD Group (*n* = 16)	*p*-Value
Male	11 (73.33%)	11 (68.75%)	0.779
Female	4 (26.67%)	5 (31.25%)	
Age (years)	60.93 ± 2.87	63.13 ± 5.75	0.241
Height (cm)	164.33 ± 8.09	163.59 ± 8.22	0.968
Weight (kg)	66.23 ± 10.85	64.34 ± 9.86	0.579
Left Foot Length (cm)	96.40 ± 5.11	94.50 ± 4.95	0.405
Right Foot Length (cm)	96.37 ± 5.13	94.59 ± 4.71	0.394
Education Level (years)	12.93 ± 3.26	13.06 ± 3.49	0.777
Montreal Cognitive Assessment Score	27.80 ± 1.42	27.81 ± 1.28	0.855

(Mean ± standard deviation) or number (%).

**Table 2 healthcare-11-00567-t002:** Clinical characteristics of Parkinson’s group.

PD Group No.	1	2	3	4	5	6	7	8	9	10	11	12	13	14	15	16
Gender	M	M	M	F	M	F	F	M	M	F	M	M	M	M	M	F
Age (yrs)	63	63	71	63	64	61	64	60	74	58	58	57	54	65	61	74
Height (cm)	161	169	169	150	168	148	155	165	155	161	172	172	170	163	176	163
Weight (kg)	75	68	63	62	80	51	57	62	58	55	80	65	73	62	72	46
Left Leg Length (cm)	93	97	97	88	93	85	91	94	88	92	102	99	98	95	102	98
Right Leg Length (cm)	92	97	98	89	93	86	92	94	87	92	102	98	98	94	101	99
Education Level (yrs)	6	14	14	12	16	14	6	16	14	12	18	14	16	16	12	9
MoCA	26	28	26	28	26	28	27	26	28	28	29	29	28	29	30	29
Modified H&Y	2.5	1.5	3	1.5	1.5	2.5	1.5	2.5	2.5	1.5	2.5	2.5	1.5	1	1	2
Disease Duration (yrs)	4	3	6	5	7	3	7	11	6	2	4	6	4	4	3	9
Levodopa Equivalent Dosage (mg/day)	600	500	770	500	600	360	850	750	710	460	400	700	500	360	180	620

**Table 3 healthcare-11-00567-t003:** The dual tasks in the designated task priority.

	Control Group (*n* = 15)	PD Group (*n* = 16)	*p*-Value
Focus on Walking Task			
Walking Speed (cm/s)	127.93 ± 14.54	120.39 ± 27.90	0.22
Step Length (cm)	63.98 ± 5.43	62.49 ± 10.99	0.519
Step Time(s)	0.50 ± 0.04	0.53 ± 0.06	0.317
CV Step Length (%)	4.12 ± 1.93	4.78 ± 2.51	0.401
CV Step Time (%)	4.43 ± 1.39%	6.25 ± 2.54	0.093
Response Accuracy (%)	95.33 ± 13.56	72.50 ± 32.51	0.037
Reaction Time (ms)	1027.29 ± 243.03	957.83 ± 347.68	0.651
Composite Score	9.86 ± 3.15	8.21 ± 4.80	0.156
Focus on Cognitive Task			
Walking Speed (cm/s)	106.77 ± 17.38	93.60 ± 17.55	0.106
Step Length (cm)	58.26 ± 6.38	52.19 ± 8.49	0.081
Step Time(s)	0.55 ± 0.05	0.56 ± 0.05	0.561
CV Step Length (%)	3.69 ± 2.24	4.29 ± 1.39	0.121
CV Step Time (%)	4.53 ± 1.05%	7.02 ± 3.08%	0.053
Response Accuracy (%)	100.00 ± 0.00	95.00 ± 14.14	0.171
Reaction Time (ms)	1171.15 ± 387.49	1033.89 ± 290.91	0.366
Composite Score	9.52 ± 3.39	9.94 ± 3.08	0.438

## Data Availability

The data presented in this study are available on request from the corresponding author.

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
