# Peer review of "The Executive-Function-Related Cognitive–Motor Dual Task Walking Performance and Task Prioritizing Effect on People with Parkinson’s Disease"

_healthcare, 2023, doi:10.3390/healthcare11040567_

Round 1

Reviewer 1 Report

Comments:

The research question is clinically important but I am not sure if it is adding any new information to the existing knowledge. The study design is sound and the manuscript is well written. Some points needs to be addressed:

1-     The differences in the outcome parameters among the patients and control are expected.

2-     The text in the first paragraph of the results section (Page 6, page 205-208) seems like instruction. Should be deleted.

3-     Page 6, line 211, correct the number for control group to 15.

4-     Inclusion criteria must be clearly defined. Stable medication usage should be properly defined.

5-     Information about medication use should be provided in the baseline characteristics.

6-     Types of PD e.g tremor predominant, Akinetic or mixed type of IPD can have different outcomes in this test. Authors must state the types of included PD patients and also discuss them in the discussion part.

7-     Clinical characteristics like duration of disease, associated non- motor features, drugs etc should be provided.

8-     Who were the assessors? Was it single or multiple assessors? Was the assessor blinded?

9-     What was the average duration for performing the tasks in both the groups?

Reviewer 2 Report

The manuscript titled “The executive function related cognitive-motor dual task walking performance and task prioritizing effect on people with Parkinson Disease” aimed to carry out an investigation on cognitive-walking dual task in patients with PD. Authors designed cognitive-motor dual task walking tests using executive functions related cognitive tasks. They evaluated the effect of task prioritization as well. Participants were 16 PD patients and 16 healthy individuals (control group). Both PD patients and healthy individuals were included in the study if their MoCA score was at least 26. Single cognitive test, including spatial memory test, visual stroop test and calculation test were administered to each participant. Furthermore, a single walking test and a dual walking test (i.e., dual spatial memory test, dual visual stroop test, dual calculation test) were administered to each participant. Main results showed that PD group performed worse than the control group in single and dual walking test. Furthermore, significant differences were found in composite score under the dual calculation walking task. Lastly, during the dual walking test, the PD group had lower response for accurate rate in cognitive test when prioritizing was assigned to walking. Authors discussed their results in light of previous literature, highlighting strengths and limitations of their study.

I carefully read the manuscript, and I think it may be of interest for the readers of Healthcare. Nevertheless, I think that it could be worth considering some points before the publication. Below there are my comments and suggestions.

Introduction

Page 1, line 42: Author(s) mentioned the prevalence of PD. Please, they should clarify which geographical area they referred to.

Page 1, lines 44-50: These sentences should be supported by appropriate references.

Page 2, line 57: Author(s) cited the dual walking test paradigm. A better explanation of it could be appropriate.

Page 2, line 77: Author(s) claimed “The dual task test paradigm could be explained by capacity theory and/or bottleneck theory”. A brief explanation about it could be appropriate.

Page 3, lines 123-127: the aims of the study should be clarified. Author(s) claimed that this study aimed to design a cognitive-walking test that used a cognitive task related to executive functions. They hypothesized that this kind of dual walking test would be sensitive to cognitive deficits for early-stage PD patients. Nevertheless, they did not performed analyses to quantify sensitivity or specificity of their cognitive walking test. I think that this point could be ambiguous. Author(s) should clarify this.

Materials and Methods

Page 3, line 138: Subjects were included in the study if their MoCA score was at least 26. Why did they choose this cut-off? Some references are needed.

Page 4, line 152: Single and dual tests were performed twice each. What does that mean? For example, the visual stroop test was performed twice? Author(s) should specify if each single and dual test were repeated and how they avoided the learning effect.

Page 4, lines 157-158: “Each task is performed twice.” How was the learning effect avoided?

Overall, Author(s) intended to employ cognitive task related to executive functions. As well-established, the visual stroop test is included in executive functions task. What about spatial memory and calculation? For example, a task of calculation is related to working memory. This point is not clear. Author(s) should specify which cognitive domain is engaged for each task.

Results

Results were well reported.

Discussion

It could be convenient to start with a description about aims and sub-aims of the study, and related results.

Page 9, lines 259-261: Author(s) indicated that their main finding was that the walking performance of PD group was significantly worse than the control group in single and dual task walking. Nevertheless, in the introduction section, Author(s) indicated that they aimed to design a cognitive-motor dual task that would be sensitive to cognitive deficits in PD patients. What about this? How did Authors reach this aim?

Overall, results concern the prioritizing effect needed to be examine in depth and better discussed in this section. 

Round 2

Reviewer 2 Report

Dear Editor,

I read the revised version of the manuscript and I think that it has improved with respect to the previous version. Authors did all the recommended changes and answered to all the questions raised by the reviewers. My suggestion is to accept it in the present form.

Best regards,